# Alterations in Intestinal Antioxidant and Immune Function and Cecal Microbiota of Laying Hens Fed on Coated Sodium Butyrate Supplemented Diets

**DOI:** 10.3390/ani12050545

**Published:** 2022-02-22

**Authors:** Sasa Miao, Zuopeng Hong, Huafeng Jian, Qianqian Xu, Yating Liu, Xiaoming Wang, Yan Li, Xinyang Dong, Xiaoting Zou

**Affiliations:** 1Key Laboratory of Animal Feed and Nutrition of Zhejiang Province, Key Laboratory of Animal Nutrition and Feed Science (Eastern of China), Ministry of Agriculture and Rural Affairs, The Key Laboratory of Molecular Animal Nutrition, Ministry of Education, College of Animal Sciences, Zhejiang University, Hangzhou 310058, China; 21917066@zju.edu.cn (S.M.); jianhuafeng@zju.edu.cn (H.J.); qianqianxu@zju.edu.cn (Q.X.); 22017064@zju.edu.cn (Y.L.); 22017083@zju.edu.cn (X.W.); 22117017@zju.edu.cn (Y.L.); sophiedxy@zju.edu.cn (X.D.); 2Jiande Weifeng Feed Co. Ltd., Jiande 311600, China; zphong0120@163.com

**Keywords:** coated sodium butyrate, immune function, cecal microbiota, laying hens

## Abstract

**Simple Summary:**

Sodium butyrate has attracted considerable attention as a promising feed additive due to its biological function on the intestine. The results of the present study indicate that coated sodium butyrate could improve intestinal health via regulating inflammatory factors, enhancing the superoxide dismutase activity, as well as total antioxidative capacity activity of the small intestine, increasing the production of short-chain fatty acids and modulating the cecum microbial communities of laying hens. To the best of our knowledge, few reports on laying hens have investigated the effects of coated sodium butyrate on gut health by influencing cecal microflora and metabolites. This study will provide an alternative strategy for enhancing the gut health of laying hens.

**Abstract:**

This study was designed to evaluate the effects of dietary coated sodium butyrate (CSB) on the intestinal antioxidant, immune function, and cecal microbiota of laying hens. A total of 720 52-week-old Huafeng laying hens were randomly allocated into five groups and fed a basal diet supplemented with CSB at levels of 0 (control), 250 (S250), 500 (S500), 750 (S750), and 1000 (S1000) mg/kg for eight weeks. The results revealed that CSB supplementation quadratically decreased the malondialdehyde content and increased the superoxide dismutase activity of the jejunum as well as the total antioxidative capacity activity of the ileum (*p* < 0.05). Dietary CSB supplementation linearly decreased the diamine oxidase and D-lactic acid content of the serum (*p* < 0.05). Compared with the control group, the addition of CSB resulted in linear and/or quadratic effects on the mRNA expression of inflammatory cytokines TNF-α, IL-6, and IL-10 in the jejunum and ileum (*p* < 0.05). The short-chain fatty acid concentrations increased quadratically as supplemental CSB improved (*p* < 0.05). Additionally, dietary CSB levels had no effect on microbial richness estimators, but ameliorated cecal microbiota by raising the abundance of probiotics and lowering pathogenic bacteria enrichment. In conclusion, our results suggest that dietary supplementation with CSB could improve the intestinal health of laying hens via positively influencing the antioxidant capacity, inflammatory cytokines, short-chain fatty acids, and gut microbiota. In this study, 500 mg/kg CSB is the optimal supplement concentration in the hens’ diet.

## 1. Introduction

The intestine acts as a dynamic system consisting of interactions among the gut flora, epithelial barrier, and immune cells [1]. The intestines of animals are continuously exposed to various antigens, such as food-driven antigens and microbes [2,3], which could challenge their health status. A number of investigations have emphasized the necessity of supporting feed supplements that maintain gut health [4,5]. Sodium butyrate as a green feed additive attracts considerable attention on account of its biological function on the intestines in livestock [6].

Butyrate is generated by the digestion of carbohydrates and proteins by beneficial bacteria in the latter intestine [7] and is the main regulator of cell proliferation and differentiation [8]. Studies have proved that sodium butyrate could not merely strengthen the function of intestinal defense and effectively ameliorate oxidative stress, but also has potential immunomodulatory and anti-inflammatory properties in the gut [9,10]. Moreover, butyrate also plays a pivotal role in elevating intestinal integrity and maintaining the balance of the microbial ecosystem [11]. As recently reported in turbot fed with high-soybean meal diets, sodium butyrate could improve the intestinal barrier function and restore the intestinal flora imbalance by altering the abundance of *Proteobacteria*, *Bacteroidetes*, *Deinococcus-Thermus,* and *Actinobacteria* [1]. The study by Yang et al. [12] also demonstrated that butyrate glycerides can promote chicken health and well-being via altering intestinal microbiota composition.

Intestinal microbes mainly exist in the cecum of poultry. There are multitudes of evidence revealing that a close correlation exists between intestinal microbes and the productive performance of poultry [13,14]. Short-chain fatty acids (SCFAs) are deemed as one of the richest metabolites of intestinal microorganisms and effectively regulate the microbial balance and maintain the stability of the internal environment [15]. Nevertheless, there remain rare reports on the effects of sodium butyrate supplementation on the cecal microbiota and metabolites of laying hens. Due to the irritant odor and low utilization in the gut of sodium butyrate [16], we carried out this experiment with coated sodium butyrate (CSB). Therefore, this trial was conducted to evaluate the effects of CSB addition on the intestinal antioxidant enzyme, immunity, and cecal SCFAs and microbiota in laying hens, which may be a reference on elucidating the regulatory mechanism for improving the intestinal health of laying hens.

## 2. Materials and Methods

All experimental protocols involving animals were approved by the Animal Care and Welfare Committee of Animal Science College and the Scientific Ethical Committee of Zhejiang University (No. ZJU2013105002) (Hangzhou, China).

### 2.1. Experimental Design, Animals, and Diet

A total of 720 52-week-old Huafeng laying hens at late laying period (initial laying rate = 73.6 ± 0.3%) were arbitrarily allotted to five treatments [17], each of which comprised six replicates (*n* = 24 laying hens). The specific protocol of the groups was as follows: 1. basal diet (control); 2. basal diet + 250 mg CSB/kg diet (S250); 3. basal diet + 500 mg CSB/kg diet (S500); 4. basal diet + 750 mg CSB/kg diet (S750); and 5. basal diet + 1000 mg CSB/kg diet (S1000). The CSB (coated with palm oil and silica, including 50% sodium butyrate) was supplied by Hangzhou Dade Biotechnology Co. LTD (Hangzhou, China). During the whole trial period (9 weeks, covering a 1-wk acclimation stage and an 8-wk test period), laying hens were raised in the poultry house with the temperature maintained at approximately 23 °C and a light regime lasting 16 h/day. The water and diets were freely accessed by the laying hens throughout the trial period. The composition of the basal diet is presented in Table 1.

### 2.2. Sample Collection

After fasting for 12 h, twelve birds in each treatment (two birds per repeat) were chosen arbitrarily in the eighth week of the trial period. A blood sample was collected from the wing vein and then centrifuged (3000× *g* for 15 min). The supernatant of blood was removed and immediately stored at −80 °C. The birds were then euthanized by cervical dislocation. Two segments (2 cm) of the jejunum and two segments of ileum were excised and stored at −80 °C. The cecas of six hens in each group were collected and frozen in liquid nitrogen for SCFAs and microbial composition assays.

### 2.3. Intestine and Serum Parameters Measurements

A certain amount of jejunum and ileum tissue and phosphate buffer (PBS) in a ratio of 1:9 (weight: volume) were taken to make a 10% homogenate using a tissue homogenizer, then centrifuged at 3000× *g* for 10 min to separate the supernatant for subsequent determination. The activities of total antioxidative capacity (T-AOC), catalase (CAT), superoxide dismutase (SOD), and malondialdehyde (MDA) in the intestinal supernatant, as well as the levels of Diamine oxidase (DAO), D-lactic acid (DL), and total bile acid (TBA) in the serum, were measured and calculated according to the kit procedure (Nanjing Jiancheng Bioengineering Institute, Nanjing, China).

### 2.4. Total RNA Isolation, and Quantitative Real-Time PCR (qRT-PCR)

The detailed methods of total RNA isolation and qRT-PCR are the same as described previously [17,18]. Gene-specific primers for tumor necrosis factor-alpha (TNF-α), interleukin-1 beta (IL-1β), interleukin-6 (IL-6), interleukin-10 (IL-10), and β-Actin are summarized in Table 2. The β-Actin served as an endogenous gene. The relative mRNA expression levels of the genes were counted by the 2^−ΔΔCt^ method.

### 2.5. Cecal SCFAs Concentration Analysis

Six samples from each group were used to determine the concentrations of SCFAs using the protocol of SCFAs analysis based on a previous study [19]. In brief, 300 mg of chyme was weighed and added with 3 mL of a sterile phosphate buffer (PBS). After vortex oscillation, the solution was centrifuged for 10 min at 4 °C (12,000× *g*); 1000 µL of supernatant was taken and added to 200 µL of 25% metaphosphoric acid, oscillated and shaken well. The sample was then placed on ice for at least 30 min and centrifuged at 12,000× *g* (10 min); finally, the separated supernatant was percolated via the syringe filter (0.22 µm) and added to the Shimadzu GC-2010 ATF equipment for the measurement of SCFAs content. N_2_ was the carrier gas (flow, 18 mL per min and pressure, 12.5 Mpa), the detector and injector temperature was 180 °C, and the temperature of the column was gradually increased from 80 to 170 °C at a speed of 4 °C per minute.

### 2.6. DNA Extraction, 16S rRNA Gene Sequencing and Data Analysis

The total bacterial DNA extraction of the cecal digesta of six laying hens from each treatment was performed with the QIAamp DNA Stool Mini Kit (Mo Bio Laboratories, San Diego, CA, USA). The fineness and quality of the DNA extracted from cecal samples were assessed with a NanoDrop ND-2000 spectrophotometer (Thermo Fisher Scientific, Wilmington, DE, USA). The V4 hyper-variable region of the 16S rRNA of all samples was magnified by the gene-specific primers (515F: GTGCCAGCMGCCGCGGTAA 806R: GGACTACHVGGGTWTCTAAT). The cleaned sequences belonged to the same operational taxonomic units (OTUs) based on a 97% similarity level. The indexes containing Coverage, Shannon, Simpson, Chao, and Sobs were used to evaluate the alpha diversity. The beta diversity based on OTU level was performed by an unweighted UniFrac principal coordinates analysis (PCoA) and principal component analysis (PCA) to distinguish between different groups. Nonmetric dimensional scaling (NMDS) was used to evaluate the difference in community structure of each sample. The data generated were analyzed on the RealBio Cloud Platform (http://cloud.Majorbio.com/, accessed on 7 January 2022).

### 2.7. Statistical Analysis

The data were statistically analyzed by one-way ANOVA followed by LSD’s multiple comparison tests or Student’s *t*-test with Welch’s correction with SPSS 20.0. Linear and quadratic effects were considered significant at *p* < 0.05. Results were expressed as mean ± SEM. Additionally, the relations between SCFAs and key parameters were explored by Spearman’s correlation analysis. Clustering correlation heatmap and network with signs were performed using the OmicStudio tools at https://www.omicstudio.cn (accessed on 7 January 2022).

## 3. Results

### 3.1. Intestinal Oxidation Status

As shown in Figure 1, in the jejunum, the activity of SOD showed an increase in a quadratic manner and reverse evidence of the MDA effect. In the ileum, T-AOC activity presented a quadratic increase as the dietary CSB addition levels improved. No difference in other antioxidant indexes was observed between the control and treatment groups.

### 3.2. Intestinal Barrier Function and Gene Expression of Inflammatory Cytokines

As exhibited in Figure 2, in the serum, dietary CSB treatment did not change the content of TBA, but markedly decreased the DL and DAO content in a linear and quadratic manner; as shown in Figure 3, the mRNA expressions of TNF-α in the jejunum were quadratically decreased as the dietary CSB levels improved. The mRNA expressions of IL-6 in the jejunum and ileum showed a quadratic decrease with the increase of dietary CSB levels and reverse evidence of an IL-10 effect.

### 3.3. SCFAs Concentrations

As shown in Figure 4, the total acid, acetic, propionic, isobutyric, and valeric concentrations showed a quadratic increase as the dietary CSB levels improved. The butyric concentration exhibited a linear and quadratic response to the CSB supplementation levels.

### 3.4. Relationship among SCFAs, Intestinal Inflammatory Cytokines, and Antioxidant Indexes

The potential association among cecal SCFAs concentration, inflammatory cytokines, and antioxidant indexes in jejunum and ileum was explored. As shown in Figure 5a,b, in the jejunum, propionate, isobutyrate, and total acid were positively related with SOD and were inversely related with TNF-α and MDA; Butyrate is positively related with IL-10 and inversely correlated with IL-6. In the ileum (Figure 5c,d), acetate, propionate, valerate, and total acid is negatively correlated with IL-6, among which propionate is positively correlated with CAT; moreover, butyrate is inversely correlated with IL-1β and positively related with IL-10, whereas isobutyrate is positively correlated with T-AOC.

### 3.5. Microbial Composition

The control and S500 groups that had a significant influence on laying rate [17] were chosen to detect cecal microbiota differences. We calculated the alpha-diversity to further assess the change of cecum microbiota during CSB induction. The value of the coverage index reached 1, indicating that the sequencing depth was sufficient (Table 3). Moreover, CSB supplementation had no effect on the richness indices (chao) as well as estimator diversity (simpson and shannon). As shown in Figure 6, the results revealed that the control and S500 groups are not absolutely separated. Therefore, the integrated results suggest that CSB treatment did not alter the α and β diversity of cecum microbiota.

As exhibited in Figure 7a,c, CSB supplementation markedly altered the microbiota composition of phylum, class, and family levels. CSB inclusion remarkably reduced the abundances of *Bacteroidota* and elevated the *Firmicutes* at the phylum level (Figure 7a). CSB inclusion remarkably reduced the abundances of *Bacteroidia* and elevated the *Clostridia* at the class level (Figure 7b). CSB inclusion remarkably elevated the abundances of *Lachnospiraceae* and *Oscillospiraceae* and reduced the *Prevotellaceae* and *Bacteroidaceae* at the family level (Figure 7c).

Significant differences in the microbial compositions between the control and S500 group were further identified using linear discriminant analysis (LDA) combined with effect size (LEfSe). LDA and LEfSe analyses were used to differentiate the gathering of specific bacterial taxa whose abundance varied with dietary CSB levels and thus could serve as biomarkers. Figure 8 displayed the species with significant differences with an LDA score > 2.0. Results showed *Prevotellaceae* as a biomarker of the control group, while *Clostridia*, *Lachnospirales*, *Oscillospirales,* and *Gastranaerophilales*, which are known to SCFAs producers, were biomarkers of the S500 group.

## 4. Discussion

Butyrate is mostly produced via microbial fermentation of dietary fibers in the latter intestinal tract [7]. Butyrate has received particular focus for its beneficial function on gut homeostasis and energy metabolism [8]. In this trial, we assessed the effects of dietary CSB levels on the intestinal antioxidant, immune function, and cecal microbiota of laying hens. We found that dietary treatment with CSB impacted SCFAs and the bacterial community in favor of regulating the intestinal antioxidant and immune capacity.

The metabolism of various cells in the body produces a large number of reactive oxygen species and other substances. Laying hens are susceptible to producing excessive reactive oxygen species leading to intestinal oxidative stress due to the genetic selection for long-laying eggs [20]. Sufficient studies have proved that butyrate performed powerful antioxidant functions whether in vivo or in vitro [21,22,23,24]. Here, the intestinal antioxidant status was assessed via the monitoring of several antioxidant-related enzymes. The activities of SOD and CAT are considered as the first-line of defense in scavenging free radicals to protect cells from oxidative damage [25]. The ability of the non-enzymatic antioxidant defense system is generally measured by T-AOC [26]. MDA is a reliable indicator of lipid peroxidation [27]. As shown in our results, CSB supplementation not merely elevated SOD activity in the jejunum but also increased T-AOC activities in the ileum. Our results indicate that CSB inclusion could boost the antioxidant defense system via enzymatic and non-enzymatic antioxidants. Moreover, we found that CSB treatment decreased MDA content in the jejunum, demonstrating that CSB could prevent lipid peroxidation in the intestinal tract of laying hens. These integrated results validated that CSB has a positive improvement on the intestinal antioxidant system of laying hens.

Intestinal integrity is critical for intestinal homeostasis. DAO and DL are employed as a circulatory index of reflecting intestinal permeability [28]. Nevertheless, there is no study on whether sodium butyrate affects the intestinal barrier function of laying hens. Our results revealed that dietary CSB supplementation notably decreased the level of DAO and DL, indicating the improvement of intestinal permeability. Intestinal permeability is closely related to cytokines [29]. Intestinal epithelial barrier integrity can be facilitated or restricted by cytokines, which can also change intestinal epithelial permeability [30,31]. The results of previously published literature has stated that butyrate exerted a powerful effect on modulating immune systems via stimulating the release of inflammatory cytokines [32]. In the present study, a similar effect is exerted by the dietary treatment of CSB, which could significantly suppress the intestinal gene expression levels of pro-inflammatory cytokine TNF-α and IL-6 and elevate the gene expression of anti-inflammatory cytokine IL-10. TNF-α and IL-6 have been regarded as contributors to gut damage and are enriched in the inflamed gut [33]. The advantageous effects of IL-10 for the maintenance of intestinal permeability have been demonstrated by multiple studies [34,35]. IL-10 has also been proven to promote intestinal epithelial cell proliferation and guard the intestinal mechanical barrier [36,37]. Based on these results, we deduced that the improvement of intestinal permeability may be ascribed to the positive function of CSB on intestinal cytokine tuning.

Intestinal homeostasis is closely associated with multitudinous microbes and their products [8,38]. SCFAs are known as one of the richest metabolites produced by gut microbes, principally containing acetate, propionate, and butyrate [39]. The increase of SCFAs could result in an appropriate PH reduction of the intestine and inhibit the reproduction of noxious bacteria and contribute to intestinal health and function [40]. Our study first confirmed that the concentration of acetate, propionate, butyrate, and total acid increased markedly to different extents due to the CSB inclusion, implying that CSB could maintain the gut at a relatively healthy level. Moreover, SCFAs modulate host biological responses, including gut integrity, lipid metabolism, and the immune system [41,42]. As the preferred energy source for enteric epithelial cells, butyrate modulates several epithelial processes throughout the gastrointestinal tract, including oxidation [43], inflammation, and barrier integrity [44]. Acetate and propionic acid are also modulators of immunity and redox signaling [45,46]. Additionally, our spearman correlation analysis showed that the SCFAs are positively related to SOD, CAT, T-AOC, and IL-10 and inversely related to pro-inflammatory cytokines. Thus, the observed change in antioxidant indexes and cytokines may be attributed to the improvement of SCFA production, which would elevate gut immunity and function.

To elucidate whether the effect of CSB on cecal SCFAs can be explained at the molecular level, we assayed the gut microbiome. The result of our 16S sequencing revealed that the dietary CSB levels had no effect on α and β diversity of cecum microbiota, which was probably due to the fact that the butyrate was mostly assimilated in the small intestine. In the current study, CSB inclusion significantly increased the abundance of *Firmicutes* and *Clostridia* (phylum *Firmicutes*) and decreased the abundance of *Bacteroidota*. *Firmicutes* and *Bacteroidota* were the chief phyla in the intestinal microbiotas of chicken [47] which mainly fermented complex polysaccharides that were not apt to digest [48]. It was discovered that SCFAs are produced mainly in *Firmicutes* [49] and butyrate suppliers primarily in *Clostridia* [50]. In our study, the cecal total acid and butyric acid concentration in the S500 group was higher than the control group, which might be in connection with the improvement of the relative abundance of *Firmicutes* and *Clostridia*. Likewise, our LEFSe results also found that the other SCFAs producers such as *Oscillospirales*, *Lachnospiraceae,* and *Gastranaerophilales* were enriched in the CSB group [51,52,53] and could explain the increase in other acids. Nevertheless, we found that *Prevotellaceae* were significantly enriched in the control group. It has been documented that *Prevotellaceae* are prominently representative bacteria in samples from inflammatory bowel disease (IBD) patients [54,55]. *Provetellaceae* is also a potential pathogen that exists in the gut environment of laying hens [56]. *Provetellaceae* could destroy the mucosal barrier function via producing sulfatases [57]. In our trial, S500 treatment significantly decreased the abundance of *Provetellaceae*, indicating that coated sodium butyrate can inhibit the growth of pathogens. The above results state that CSB supplementation could ameliorate the cecal flora composition via elevating the abundance of probiotics and SCFAs while suppressing pathogenic bacteria enrichment. In the current trial, we proved for the first time that CSB could improve gut health by regulating the cecal microflora and SCFA concentration in laying hens. In addition, our study has shown that CSB has a positive effect on elevating the intestine antioxidant capacity, however, the specific mechanisms have yet to be fully illustrated.

## 5. Conclusions

In conclusion, dietary coated sodium butyrate has positive roles in gut health via regulating inflammatory factors, enhancing antioxidant capacity, increasing SCFA production, and modulating the cecum microbial communities, thus providing an alternative tactic for enhancing the intestinal health of laying hens.

## Figures and Tables

**Figure 1 animals-12-00545-f001:**
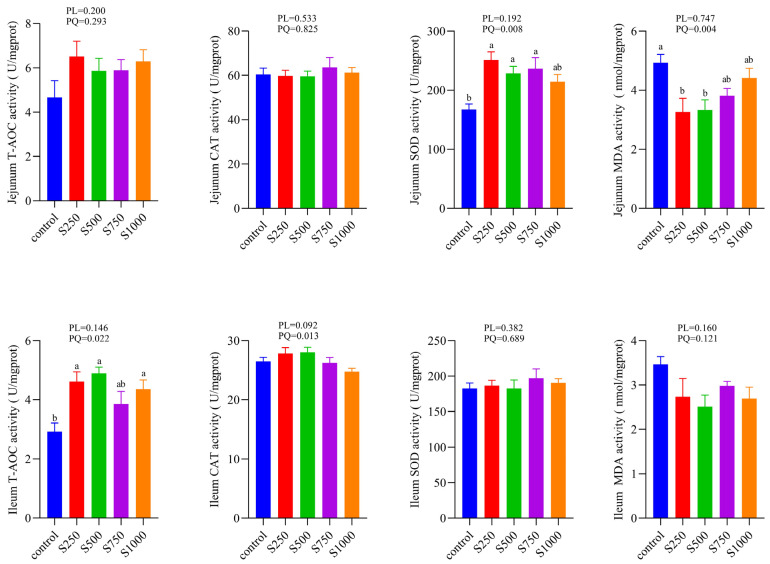
Effects of dietary CSB treatment on antioxidase changes in jejunum and ileum (*n* = 6). ^a,b^ Means with different superscripts within a column differ significantly (*p* < 0.05). T-AOC: total antioxidant capacity; CAT: catalase; SOD: superoxide dismutase; MDA: malondialdehyde.

**Figure 2 animals-12-00545-f002:**
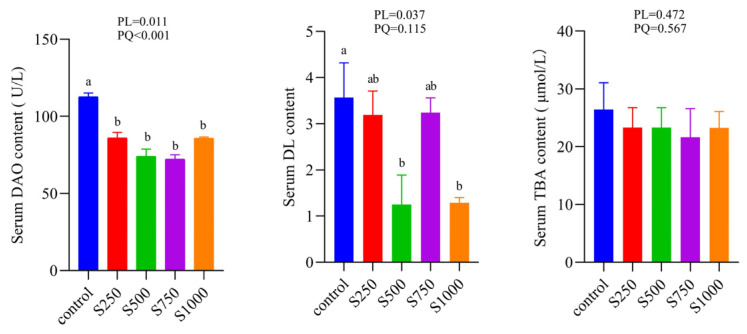
Effects of dietary CSB levels on serum biochemical indices (*n* = 6). ^a,b^ Means without common letters above the histogram differ significantly (*p* < 0.05). DAO: Diamine oxidase, DL: D-lactic acid, TBA: Total bile acid.

**Figure 3 animals-12-00545-f003:**
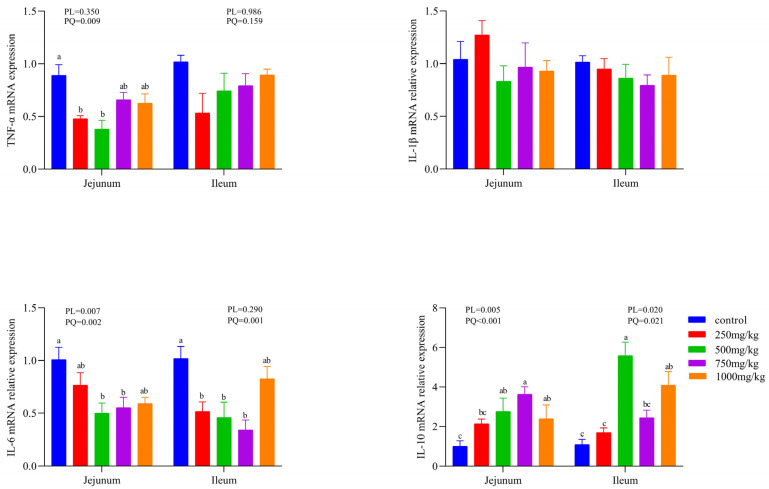
mRNA expression levels of inflammatory cytokines in the jejunum and ileum of laying hens (*n* = 6). ^a–c^ Means with different superscripts within a column differ significantly (*p* < 0.05). TNF-α: Tumor necrosis factor-alpha; IL-1β: interleukin-1 beta; IL-6: interleukin-6; IL-10: interleukin-10.

**Figure 4 animals-12-00545-f004:**
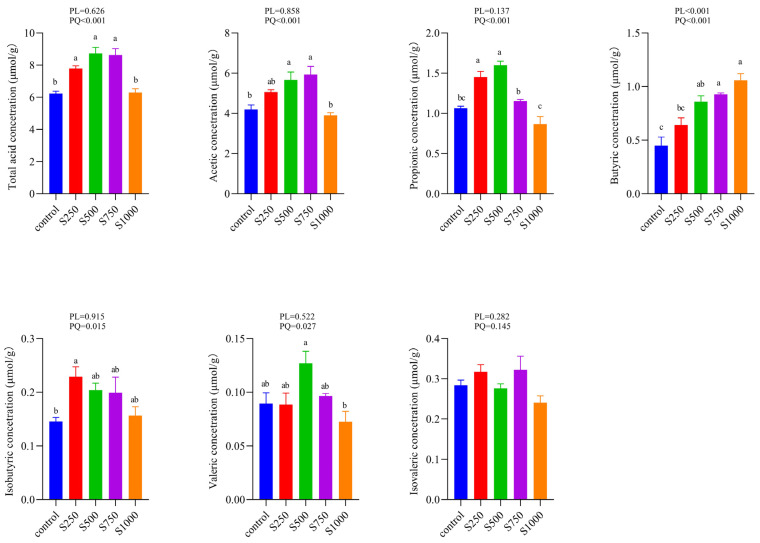
Cecum SCFAs contents of laying hens (*n* = 6). ^a–c^ Means with different superscripts within a column differ significantly (*p* < 0.05).

**Figure 5 animals-12-00545-f005:**
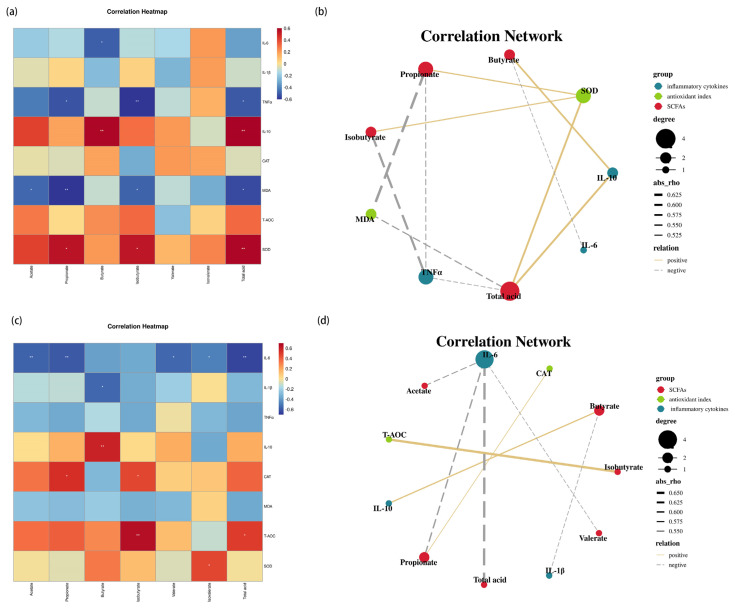
Correlation heatmap (**a**) and Correlation network (**b**) between SCFAs and inflammatory cytokines, and antioxidant indexes in the jejunum; Correlation heatmap (**c**) and Correlation network (**d**) between SCFAs and inflammatory cytokines, and antioxidant indexes in the ileum. * *p* < 0.05 and ** *p* < 0.01.

**Figure 6 animals-12-00545-f006:**
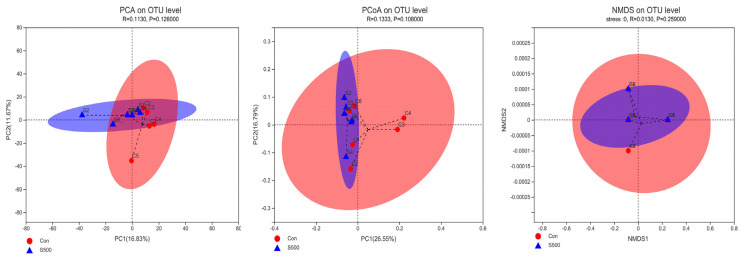
The microbial beta diversity was assessed by principal component analysis (PCA), principal coordinate analysis (PCoA), and nonmetric multidimensional scaling (NMDS) analysis based on the OTU table.

**Figure 7 animals-12-00545-f007:**
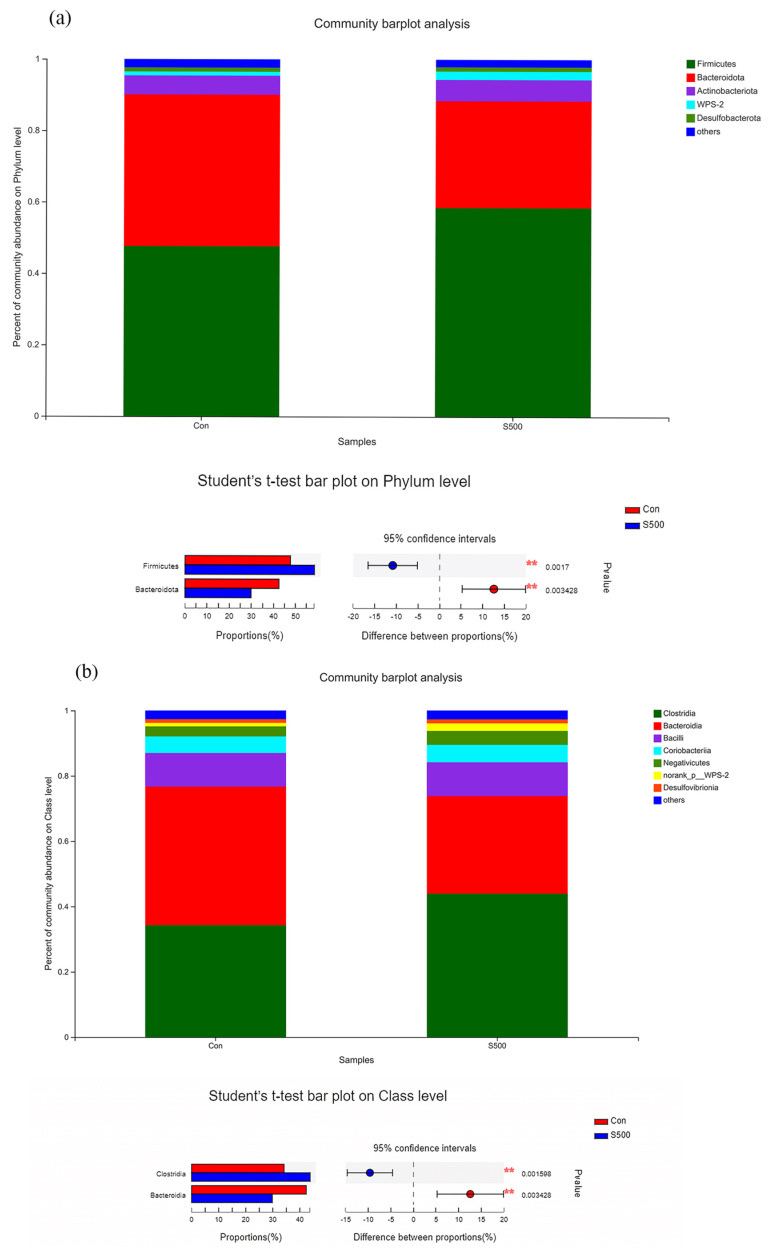
Microbial composition and significant differential bacteria in the cecum of laying hens at the phylum level (**a**), class level (**b**), and family level (**c**). Statistical differences between the two groups were calculated by Student’s *t*-test with Welch’s correction. * *p* < 0.05 and ** *p* < 0.01.

**Figure 8 animals-12-00545-f008:**
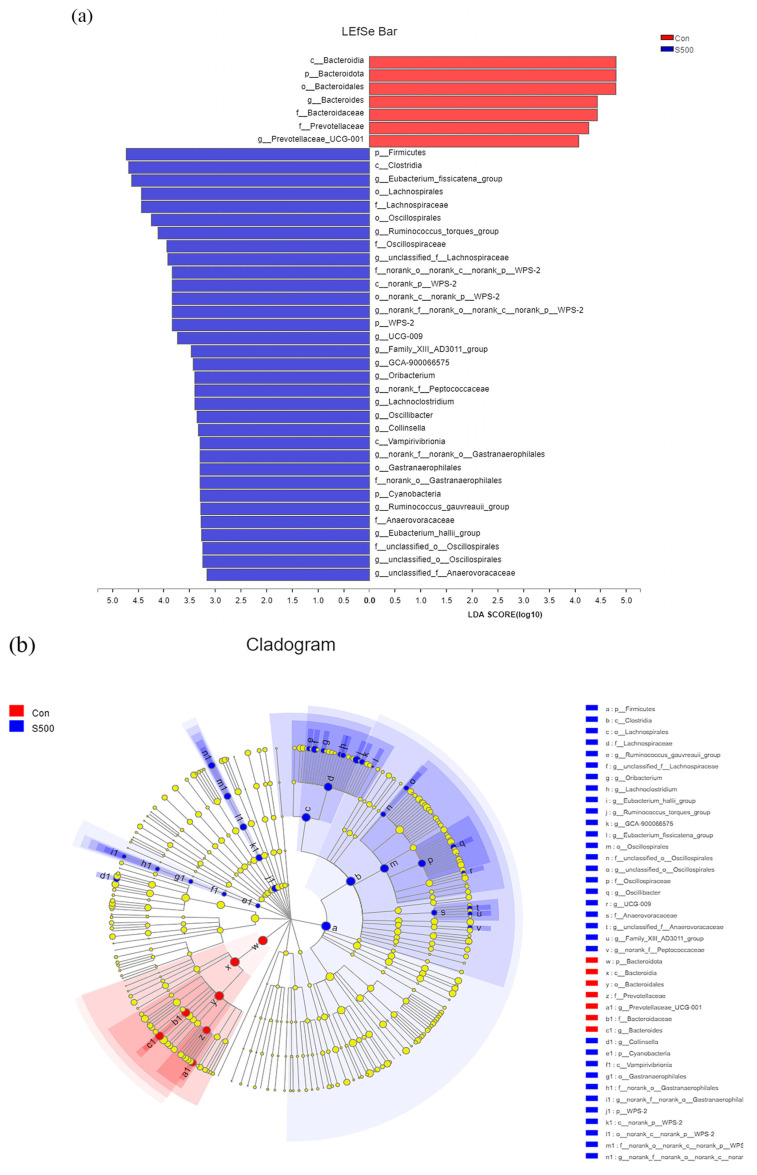
LEfSe (**a**) and LDA (**b**) analyses based on OTUs characterized the microbiomes of the control and S500 groups.

**Table 1 animals-12-00545-t001:** Ingredient compositions and nutrient levels of basal diet for hens.

Ingredients	Value	Nutrient Level ^3^	Value
Corn, %	62	Metabolism energy, MJ/kg	10.99
Soybean meal, %	24.5	Crude protein, %	15.67
Soybean oil, %	0.5	Lysine, %	0.80
Limestone, %	8	Methionine, %	0.34
Premix ^1,2^, %	5	Calcium, %	3.69
Total	100	Total phosphorus, %	0.54

^1^ Supplied vitamin and mineral per kilogram of diet: V_A_ 7500 IU, V_D3_ 2500 IU, V_E_ 49.5 mg, V_K3_ 2.5 mg, V_B__1_ 1.5 mg, V_B2_ 4 mg, V_B6_ 2 mg, V_B12_ 0.02 mg, Sodium chloride 2500 mg, chloride choline 400 mg, biotin 0.16 mg, pantothenic acid 10 mg, folic acid 1.1 mg, niacin 30 mg, Zn 80 mg, Mn 60 mg, Cu 20 mg, Fe 80 mg, I 0.8 mg, Se 0.3 mg. ^2^ Supplied per kilogram of diet in 5 groups: 0, 250, 500, 750 and 1000 mg coated sodium butyrate. ^3^ Values were calculated from data supplied by the feed database in China.

**Table 2 animals-12-00545-t002:** Primer used for real-time quantitative fluorescence PCR analysis.

Target Gene	Primer	Primer Sequence (5′-3′)	Product Length	Accession No.
β-Actin	Forward	TCCCTGGAGAAGAGCTATGAA	113 bp	NM_205518.1
	Reverse	CAGGACTCCATACCCAAGAAAG		
IL-10	Forward	CCAGGGACGATGAACTTAACA	251 bp	NM_001004414.2
	Reverse	GATGGCTTTGCTCCTCTTCT		
IL-1β	Forward	CTTCACCCTCAGCTTTCACG	137 bp	XM_015297469.2
	Reverse	CCCTCCCATCCTTACCTTCT		
IL-6	Forward	TTCAGAGTGACCTACACAGGC	146 bp	XM_015281283.2
	Reverse	GATGCTTTATCATGCGCTGC		
TNF-α	Forward	GACAGCCTATGCCAACAAGTA	244 bp	AY765397.1
	Reverse	TCCACATCTTTCAGAGCATCAA		

**Table 3 animals-12-00545-t003:** The alpha diversity of the 16S rRNA gene libraries from cecal microbiota of laying hens in the control and S500 groups (*n* = 6).

Items	Control	S500	*p*-Value
coverage	1.00	1.00	0.590
chao	934.67	959.08	0.438
sobs	809.50	839.33	0.364
shannon	4.97	5.00	0.669
simpson	0.02	0.02	0.445

## Data Availability

The data presented in this study are available on request from the corresponding author.

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
