# Peer review of "Alterations in Intestinal Antioxidant and Immune Function and Cecal Microbiota of Laying Hens Fed on Coated Sodium Butyrate Supplemented Diets"

_animals, 2022, doi:10.3390/ani12050545_

Round 1

Reviewer 1 Report

This study entitled Alterations in intestinal antioxidant and immune function and cecal microbiota of laying hens fed on coated sodium butyrate supplemented diets is interesting to the reader. However, I felt that this study needs to be clearer and change all of the figures. Because the quality of all the figures is too low to be review, I recommend to the authors resubmit it after editing the figure.

Major concern

  1. All the figures especially microbiome and correlation part must change to high resolution. I cannot recognize any words and contents. So, I can’t review result part (L186-246, Figure 5-8).
  2. Explain about the CBS composition. It containing 50% sodium butyrate and how about rest of 50%?
  3. The results are not relevant to the figure. I am recommending to the author rewrite results and make them clear.
  4. Is there any law or recognition from animal ethic commitment? If authors are approved for animal experimental ethics, please mention the information such as approval number or etc.
  5. Line 201: Why did you choose S500 CSB for this study?

Minor concern

Simple summary

  1. Line 15-16: “Coated sodium butyrate could improve antioxidant properties” it is confusing; I am suggesting to write specific antioxidant properties.
  2. Line 16: What is SCFAs?

Abstract

  1. Line 34: Authors chose 500mg/kg CBS as the best dose. In conclusion, rather than simply writing that CBS is good, it would be better to mention the optimal concentration as well.

Introduction

  1. Line 42: what do you mean antigen here?

Materials and Method

  1. Line 72: Why did you choose 52 wks laying hen for your study?
  2. Line 75: I am afraid, if you provide ad libitum feed to the laying hen, the egg production must decrease due to fat deposition in the uterus.
  3. Line 79. Mention the group name like control, S250, S500, S750 and S1000 with their explanation.
  4. Line 92. ‘chosen’. What is the selection criteria? Isn’t it randomly selected?
  5. Line 92-93: Correct the information order. First will be fasting then select the birds and finally collect blood.
  6. Line 97: Which birds did you select for cecas, control or treated birds? write the protocol of SCFAs and microbial analysis.
  7. Line 100. How to extract the sample from intestine? Please mention about the preparation method.
  8. Line 101-106. Please re-order according to result. M&M showed serum analysis first, then antioxidant enzymes. But Result showed antioxidant enzymes from intestine then serum factors.
  9. Line 104. Malondialdehyde > malondialdehyde
  10. Line 109: The gene name should be italic throughout the manuscript.
  11. Line 112. How to quantify the PCR results?
  12. Line 116: what is r/min. Author should mention xg instead of RPM
  13. Line 137. Check the ‘space’ within the link.
  14. Why the authors analysed both linear and quadratic effects in this study? In results parts mentioned just ‘there have linear effect or quadratic effects’.

Results

  1. Line 167: Please explain why serum DL decrease in 500 but again increase 750 and again decrease1000
  2. Line 169: What is coversely?
  3. Line 170-171: There is no mention of Tnf-a and IL-10 in the figure description.
  4. Lines 170, 182. jejunum or ileum. Linear or quadratic manner. Why there used ‘or’ word?
  5. Line 208: How did you say that the CSB treatment did not change the structure of the microbial composition?
  6. Line 242: The result of Figure 8 needs more explanation.

Discussion

  1. Line 251: How could you simply write CSB effect cecal metabolism?
  2. Line 254: what are intestinal antioxidant properties?

Conclusions

  1. Authors want to check the gut health using inflammatory factor, antioxidant properties etc. But, the method of confirming gut health through histological analysis of the small intestine is the most accurate.

Table and Figure legends

  1. Table 1 (L87). ‘the premix in 6 treatments’. Why there written ‘6 treatments’? there are 5 treatments in this manuscript (control, 250, 500, 750, 1000).
  2. Table 1: What is the difference between Items and Ingredients as well as Composition and content?
  3. Table 2. What is the second ‘Reverse sequence’ in b-Actin?
  4. I am recommending for add the product length of all primers in table 2.
  5. Figure 1, 2, 3, 4. ‘(n=6)’. Please check this number. In M&M, there are 12 birds for each groups and there are only 5 groups.
  6. Figure 1. Please re-order the abbreviations according to figure.

Figure 3. Each abbreviations of gene name need to full name in the footnote.

Author Response

Thank you for your comments concerning our manuscript. Any change to our manuscript within the document following your comments was highlighted by using the Red Colored Text.

Major concern

1. All the figures especially microbiome and correlation part must change to high resolution. I cannot recognize any words and contents. So, I can’t review result part (L186-246, Figure 5-8).

Answer: Thank you for your comments. We apologize for our unclear figure. We have changed all the figures with high resolution (Figure 1-Figure 8).

2. Explain about the CBS composition. It containing 50% sodium butyrate and how about rest of 50%?

Answer: Thank you for your comments. The remaining 50% of CSB is the coating material, consisting of palm oil and silica. And we added the composition to the “Material and Method” in the manuscript (Line 82-83). 

3. The results are not relevant to the figure. I am recommending to the author rewrite results and make them clear.

Answer: Thank you for your comments.We have reformulated our results to make them more relevant and clear (Line 157-159; Line 167-172; Line 181-184; Line 189-192; Line 207-212; Line 219-224; Line 243-250).

4. Is there any law or recognition from animal ethic commitment? If authors are approved for animal experimental ethics, please mention the information such as approval number or etc.

Answer: Thank you for your comments. The experimental procedures were approved by the Animal Care and Welfare Committee of Animal Science College and the Scientific Ethical Committee of Zhejiang university. We have offered the ethical approval file to the assistant Editor earlier. And now we added the approval number in the manuscript (Line 74-76).

5. Line 201: Why did you choose S500 CSB for this study?

Answer: There are several studies (Angelakis et al, 2017; Wang et al, 2020) revealing that a close correlation existed between intestinal microbes and productive performance of poultry (Line 61-63). In our previously published paper (Miao et al, 2021), we found significant differences in laying rate only between the control group and S500 group (Line 204-205). We speculated that there might be differences in intestinal flora, so the control and S500 group were selected for 16S detection.

Reference:

[1]Angelakis, E. Weight gain by gut microbiota manipulation in productive animals. Microb. Pathog. 2017, 106, 162-170.  https://doi: 10.1016/j.micpath.2016.11.002.

[2]Wang, Y.; Xu, L.; Sun, X.; Wan, X.; Sun, G.; Jiang, R.; Li, W.; Tian, Y.; Liu, X.; Kang, X. Characteristics of the fecal microbiota of high- and low-yield hens and effects of fecal microbiota transplantation on egg production performance. Res. Vet. Sci. 2020, 129, 164-173. https://doi: 10.1016/j.rvsc.2020.01.020.

[3]Miao, S.S.; Zhou, W.T.; Li, H. Y.; Zhu, M. K.; Dong, X. Y.; Zou, X. T. Effects of coated sodium butyrate on production performance, egg quality, serum biochemistry, digestive enzyme activity, and intestinal health of laying hens. Ital. J. Anim. Sci. 2021, 38320, 1452-1461. https://doi:10.1080/1828051x.2021.1960209.

Minor concern

Simple summary

6. Line 15-16: “Coated sodium butyrate could improve antioxidant properties” it is confusing; I am suggesting to write specific antioxidant properties.

Answer: Thank you for your comments. We have supplemented the specific antioxidant properties in the manuscript (Line 16).

7. Line 16: What is SCFAs?

Answer: Thank you for your comments. The meaning of SCFAs is the short-chain fatty acids. We have substituted “short-chain fatty acids” for SCFAs in the manuscript (Line 17).

Abstract

8. Line 34: Authors chose 500mg/kg CBS as the best dose. In conclusion, rather than simply writing that CBS is good, it would be better to mention the optimal concentration as well.

Answer: Thank you for your comments. We have revised it according to your suggestion (Line 38).

Introduction

9. Line 42: what do you mean antigen here?

Answer: Thank you for your comments. Antigen is for the foreign proteins derived from ingested food and commensal microorganisms.

Materials and Method

10. Line 72: Why did you choose 52 wks laying hen for your study?

Answer:With the increase of age, metabolic disease intensification in the later stage of laying is a bottleneck problem to be solved.The nutritional regulation of intestinal health of laying hens in late laying period is urgent extremely.In this study,the laying rate of huafeng laying hens at 52 weeks is about 73%, which is in the late stage of laying.Therefore, 52 week-old laying hens were selected to explore the nutritional regulation of coated sodium butyrate on intestinal health of laying hens.   

11. Line 75: I am afraid, if you provide ad libitum feed to the laying hen, the egg production must decrease due to fat deposition in the uterus.

Answer: Thank you for your comments.We are sorry for our unclear expression. The specific amount of feed is not unfounded. In the acclimation stage, we will count the daily feed requirements of laying hens. We fed laying hens twice a day based on the data and made small adjustments on a case-by-case basis rather than massive changes.And we have reformulated the sentence to make clear expression (Line 87-88).

12. Line 79. Mention the group name like control, S250, S500, S750 and S1000 with their explanation.

Answer: Thank you for your comments. We have added the explanation of group name in the manuscript according to your suggestion (Line 80-83).

13. Line 92. ‘chosen’. What is the selection criteria? Isn’t it randomly selected?

Answer: Thank you for your comments. We have substituted “chose’ for “chosen”. The hens were randomly chose in this study. We have revised it (Line 97-98).

14. Line 92-93: Correct the information order. First will be fasting then select the birds and finally collect blood.

Answer: Thank you for your comments. We have corrected the information order (Line 97-99).

15. Line 97: Which birds did you select for cecas, control or treated birds? write the protocol of SCFAs and microbial analysis.

Answer: Thank you for your comments. The cecas of six hens in each group were collected at random. We have written the the protocol of SCFAs and microbial analysis (Line 102; Line 120 ; Line 146).

16. Line 100. How to extract the sample from intestine? Please mention about the preparation method.

Answer: A certain amount of jejunum and ileum tissue and phosphate buffer (PBS) in a ratio of 1:9 (weigh: volume) were taken to make into 10% homogenate using a tissue homogenizer, then centrifuged at 3000x g for ten minute to separate supernatant for subsequent determination. We have made changes to this part in the revised manuscript (Line 104-108).

17. Line 101-106. Please re-order according to result. M&M showed serum analysis first, then antioxidant enzymes. But Result showed antioxidant enzymes from intestine then serum factors.

Answer: Thank you for your comments. We have rearranged the order according to your suggestion (Line 108-112).

18. Line 104. Malondialdehyde > malondialdehyde

Answer: Thank you for your comments. We have changed the “Malondialdehyde” to “malondialdehyde” (Line 109).

19. Line 109: The gene name should be italic throughout the manuscript.

Answer: Thank you for your comments. We have checked strictly the whole manuscript and make all related correction. (Line 31; Line 116-117; Table 2; Line 169-172; Line 178-179; Line 193-196; Line 286-290).

20. Line 112. How to quantify the PCR results?

Answer: The mRNA levels were standardized as the ratio to β-actin in arbitrary units by using the 2ΔΔCt method (Line 118).

21. Line 116: what is r/min. Author should mention xg instead of RPM

Answer: Thank you for your comments. We apologize for our mistakes in writing. We have substituted “x g” for “r/min or RPM” (Line 124; Line 126).

22. Line 137. Check the ‘space’ within the link.

Answer: Thank you for your comments. We have revised it (Line 146).

23. Why the authors analysed both linear and quadratic effects in this study? In results parts mentioned just ‘there have linear effect or quadratic effects’.

Answer: Thank you for your comments.We have revised it and reformulated our results to make them more accurate (Line 157-159;Line 167-172).

Results

24. Line 167: Please explain why serum DL decrease in 500 but again increase 750 and again decrease1000.

Answer: Thank you for your comments.Serum D-lactic acid level can monitor the extent of intestinal damage and permeability. In this study,although the level of serum DL in the S750 group increased, there was no significant difference compared with other experimental groups. And compared with the control group, the level of serum DL in the S750 group showed a declined trend.Therefore, we believe that the S750 group can reduce intestinal injury and intestinal permeability, but the effect may be less obvious. In addition, in antioxidant and inflammatory factor indicators, S750 group does have a significantly positive effect. Therefore, we consider that the S750 group plays a positive role in decreasing the intestinal permeability, but the effect may be less obvious.

25. Line 169: What is coversely?

Answer: Thank you for your comments. The meaning of the “coversely” is the “ on the contrary”. We have rephrased the sentence (Line 171-172).

26. Line 170-171: There is no mention of Tnf-a and IL-10 in the figure description.

Answer: Thank you for your comments. The description of TNF-α and IL-10 in the Figure 3 has been marked red in the manuscript (Line 169-172).

27. Lines 170, 182. jejunum or ileum. Linear or quadratic manner. Why there used ‘or’ word?

Answer: Thank you for your comments. We are sorry for the inaccuracy of our expression.We have paraphrased this passage (Line 171; Line 183). 

28. Line 208: How did you say that the CSB treatment did not change the structure of the microbial composition?

Answer: Thank you for your comments.We apologize for the wrong expression. We have paraphrased this sentence (Line 210-212). 

29. Line 242: The result of Figure 8 needs more explanation.

Answer: Thank you for your comments.We have refined the results in Figure 8 (Line 243-250).

Discussion

30. Line 251: How could you simply write CSB effect cecal metabolism?

Answer: Thank you for your comments. We are sorry for the inaccuracy of our expression.We have paraphrased this sentence (Line 255-256). 

31. Line 254: what are intestinal antioxidant properties?

Answer: Thank you for your comments. Intestinal antioxidant capacity refers to the ability of the intestine to effectively inhibit the oxidation of free radicals. We have replaced the “properties” with the “capacity” (Line 258).

Conclusions

32. Authors want to check the gut health using inflammatory factor, antioxidant properties etc. But, the method of confirming gut health through histological analysis of the small intestine is the most accurate.

Answer: Thank you for your helpful comments concerning our manuscript. In our previous published paper (Miao, 2021), we have found that coated sodium butyrate had a positive effect on the morphology of small intestine. Therefore, inflammatory factors and antioxidant indexes were used in this study to further verify the effect of coated sodium butyrate on intestinal health of laying hens.

Reference

Miao, S.S.; Zhou, W.T.; Li, H. Y.; Zhu, M. K.; Dong, X. Y.; Zou, X. T. Effects of coated sodium butyrate on production performance, egg quality, serum biochemistry, digestive enzyme activity, and intestinal health of laying hens. Ital. J. Anim. Sci. 2021, 20, 1452-1461. https://doi:10.1080/1828051x.2021.1960209.

Table and Figure legends

33. Table 1 (L87). ‘the premix in 6 treatments’. Why there written ‘6 treatments’? there are 5 treatments in this manuscript (control, 250, 500, 750, 1000).

Answer: Thank you for your helpful comments concerning our manuscript. We are very sorry for this mistake caused by our negligence.We have substituted “5” for “6” (Line 94).

34. Table 1: What is the difference between Items and Ingredients as well as Composition and content?

Answer: Thank you for your helpful comments concerning our manuscript. The item refers to two parts:Ingredients (Corn, Soybean meal, Soybean oil, Limestone and Premix) and Nutrient levels ( Metabolism energy, Crude protein, Lysine, Methionine, Calcium and Total phosphorus), while Ingredients refers to Corn, Soybean meal, Soybean oil, Limestone and Premix in the feed. So are the composition and content.We have revised the table 1 to make it easier to understand (Table 1).

35. Table 2. What is the second ‘Reverse sequence’ in b-Actin?

Answer: Thank you for your helpful comments concerning our manuscript. The second ‘Reverse sequence’ in β-Actin is CAGGACTCCATACCCAAGAAAG. And we have revised in the Table 2 (Table 2).

36. I am recommending for add the product length of all primers in table 2.

Answer: Thank you for your helpful comments concerning our manuscript. We have added the product length of all primers in table 2 (Table 2).

37. Figure 1, 2, 3, 4. ‘(n=6)’. Please check this number. In M&M, there are 12 birds for each groups and there are only 5 groups.

Answer: Thank you for your helpful comments concerning our manuscript.The number (n=6) of “Figure 1, 2, 3, 4.” is correct. We refer to previous studies (Xie et al,  2019; Miao et al, 2020), two laying hens in per replicate were chose at the time of sample collection, and one laying hen in per replicate was used for index test.

Reference:

[1]Xie C, Elwan HAM, Elnesr SS, Dong XY, Zou XT. Effect of iron glycine chelate supplementation on egg quality and egg iron enrichment in laying hens. Poult Sci. 2019 Dec 1;98(12):7101-7109. doi: 10.3382/ps/pez421. PMID: 31347692.

[2]Miao L, Gong Y, Li H, Xie C, Xu Q, Dong X, Elwan HAM, Zou X. Alterations in cecal microbiota and intestinal barrier function of laying hens fed on fluoride supplemented diets. Ecotoxicol Environ Saf. 2020 Apr 15;193:110372. doi: 10.1016/j.ecoenv.2020.110372. Epub 2020 Feb 27. PMID: 32114238.

38. Figure 1. Please re-order the abbreviations according to figure.

Answer: Thank you for your helpful comments concerning our manuscript. We have re-order the abbreviations according to Figure 1 (Line 164-165).

39. Figure 3. Each abbreviations of gene name need to full name in the footnote.

Answer: Thank you for your helpful comments concerning our manuscript. We have supplemented the full name of the gene in the footnote of the Figure 3 (Line 178-179).

Reviewer 2 Report

This study investigated alterations in intestinal antioxidant and immune function and cecal microbiota of laying hens fed on coated sodium butyrate supplemented diets. The study is ethically acceptable and contains sufficient novel data to justify publication in Animals. Therefore, my decision is “accepted” for publication in Animals.

Author Response

We greatly appreciate your taking the time to review our manuscript. It is our honor to have your recognition, thank you again! 

Reviewer 3 Report

The manuscript “Alterations in intestinal antioxidant and immune function and cecal microbiota of laying hens fed on coated sodium butyrate supplemented diets” is fluently written text of an topical theme. The intestinal health in laying hens is less frequently studied than broiler gut health and thus the study increases knowledge on that field. The described study is therefore relevant to the readers of the journal. 

The manuscript is well written and easy to read. However, the authors should carefully check the use of tenses throughout the text. The authors should provide an ethical approval of the use of animals in their study. I have mainly minor comments on the manuscript. I would like the authors’ clarification regarding a few points before I can recommend this paper for publication.

- Materials and methods
o lines 77-78: The authors explain that the CSB contained 50 % of sodium butyrate. Provide more detailed information about the coating of sodium butyrate and the content of CSB (other 50% of CBS).
o lines 78-80: Was the dosage of CBS within the treatments mg/kg feed?
o line 92: How were the two birds per replicate chosen? Randomly?
o line 96: Does “Two segments (2 cm) of jejunum and ileum” mean one segment of jejunum and one segment of ileum or two segments of jejunum and two segments of ileum? Please rephrase the text.
o line 105: “detecting” refers to results and here the authors are explaining methods. A better verb is for example analyze. Rephrase the text.
o line 119: Explain the abbreviation PBS.
- Results
o line 168: It sounds a bit odd to say: “Figure 3 revealed that…”. Surely, the statistical analyses revealed the results that are presented in the Figure 3. Text needs to be revised.
o line 202: “significant difference” should be significant influence. The authors could include a short paragraph in Introduction section about the effects of intestinal microbiota on egg production.
- Discussion
o lines 248-250: The first two sentences of Discussion are lacking references. Please provide references to these sentences.
o line 255: Remove “As we all know”. This is unnecessary text.
o lines 256-256: The sentence: “ Laying hens are susceptible…” requires a reference.
o line 291: Revise “is in closely associated” to “is closely associated”
o lines 325-331: The authors suddenly refer to human studies (references 49 and 50) and to a study on pig tissues (reference 51) and make conclusions that the results apply also with laying hens. When referring to other species these species should be mentioned, and comparing the results of studies with other species should not be straightforward.
o In the discussion, a short paragraph focusing on the strengths and limitations of the study would be appreciated.

Author Response

Thank you for your comments concerning our manuscript. Any change to our manuscript within the document following your comments was highlighted by using the Blue Colored Text.

1.The manuscript is well written and easy to read. However, the authors should carefully check the use of tenses throughout the text. The authors should provide an ethical approval of the use of animals in their study. I have mainly minor comments on the manuscript. I would like the authors’ clarification regarding a few points before I can recommend this paper for publication.

Answer: Thank you for your comments. We have checked strictly the use of tenses throughout the text and make related correction. We have offered the ethical approval file to the assistant Editor earlier. And now we added the approval number in the manuscript (Line 74-76).

- Materials and methods

2.lines 77-78: The authors explain that the CSB contained 50 % of sodium butyrate. Provide more detailed information about the coating of sodium butyrate and the content of CSB (other 50% of CBS).

Answer: Thank you for your comments. The remaining 50% of CSB is the coating material, consisting of palm oil and silica. And we added the composition to the “Material and Method” in the manuscript (Line 82-83).

3.lines 78-80: Was the dosage of CBS within the treatments mg/kg feed?

Answer: Thank you for your comments. The dosage of CBS is within the treatments mg/kg feed.

4.line 92: How were the two birds per replicate chosen? Randomly?

Answer: Thank you for your comments. The hens were randomly chose in this study. And we have revised this sentence in our manuscript (Line 98).

5.line 96: Does “Two segments (2 cm) of jejunum and ileum” mean one segment of jejunum and one segment of ileum or two segments of jejunum and two segments of ileum? Please rephrase the text.

Answer: Thank you for your comments. Two segments (2 cm) of jejunum and ileum” means the two segment of jejunum and two segment of ileum. We have rephrased the text (Line 101).

6.line 105: “detecting” refers to results and here the authors are explaining methods. A better verb is for example analyze. Rephrase the text.

Answer: Thank you for your comments. We have rephrased the text (Line 111).

7.line 119: Explain the abbreviation PBS.

Answer: Thank you for your comments. PBS: phosphate buffer. We have defined the abbreviation PBS in the revised manuscript (Line 123).

- Results

8.line 168: It sounds a bit odd to say: “Figure 3 revealed that…”. Surely, the statistical analyses revealed the results that are presented in the Figure 3. Text needs to be revised.

Answer: Thank you for your comments. We have rewritten the text according to your suggestion (Line 169).

9.line 202: “significant difference” should be significant influence. The authors could include a short paragraph in Introduction section about the effects of intestinal microbiota on egg production.

Answer: Thank you for your comments. We have changed the “significant difference” to “significant influence” (Line 204). We have added a short paragraph in introduction section about the effects of intestinal microbiota on egg production (Line 61-63; Line 383-387).

- Discussion

10.lines 248-250: The first two sentences of Discussion are lacking references. Please provide references to these sentences.

Answer: Thank you for your comments. We have added the reference to these sentences (Line 253-254).

11.line 255: Remove “As we all know”. This is unnecessary text.

Answer: Thank you for your comments. We have deleted it.

12.lines 256-256: The sentence: “ Laying hens are susceptible…” requires a reference.

Answer: Thank you for your comments.We have offered the reference (Line 262; Line 401-403).

13.line 291: Revise “is in closely associated” to “is closely associated”

Answer: Thank you for your comments. We have revised it (Line 295).

14.lines 325-331: The authors suddenly refer to human studies (references 49 and 50) and to a study on pig tissues (reference 51) and make conclusions that the results apply also with laying hens. When referring to other species these species should be mentioned, and comparing the results of studies with other species should not be straightforward.

Answer: Thank you for your comments.We have rewritten the sentence to make it more coherent and added related literatures about laying hens (Line 328-334).

15.In the discussion, a short paragraph focusing on the strengths and limitations of the study would be appreciated.

Answer: Thank you for your comments.We have added a short paragraph focusing on the strengths and limitations of the study (Line 337-341).

Round 2

Reviewer 1 Report

The author corrected the reviewer's all comments. However, the author needs still some explanation before publication this manuscript.  Please check these concerns below -

  1. Line 206-207: You mention there is no difference between the control and S500 groups. Thereafter, you choose S500 for your study. But the previous study reported that in Miao et al, 2021, we found significant differences in laying rate only between the control group and S500 group. So, I recommend that you have to explain why you choose S500 based on your result.
  2. Line 43: Based on your definition. The antigen is the foreign particle. So, could you explain the difference between antigens and microbes derived from food in the intestine?
  3. Line 72: Why did you choose 52 wks laying hen for your study?

Answer: With the increase of age, metabolic disease intensification in the later stage of laying is a bottleneck problem to be solved. The nutritional regulation of intestinal health of laying hens in late laying period is urgent extremely. In this study, the laying rate of huafeng laying hens at 52 weeks is about 73%, which is in the late stage of laying. Therefore, 52 week-old laying hens were selected to explore the nutritional regulation of coated sodium butyrate on intestinal health of laying hens.

This condition you have to mention in the materials and method section with citation.

Author Response

Thank you for your comments concerning our manuscript. Those comments are very valuable and helpful for revising our paper and guiding our research. We have studied those comments carefully and have made corrections which we hope meet with approval. Any change to our manuscript within the document following your comments was highlighted by using the Orange Colored Text.

 1.Line 206-207: You mention there is no difference between the control and S500 groups. Thereafter, you choose S500 for your study. But the previous study reported that in Miao et al, 2021, we found significant differences in laying rate only between the control group and S500 group. So, I recommend that you have to explain why you choose S500 based on your result.

Answer: Thank you for your comments. In line 206-207 of our revised manuscript,  we have mentioned that the control and S500 groups that have an significant influence in laying rate rather than there is no difference (as shown in the word). We have substituted “a” for “an” (Line 205) and we are very sorry for this mistake for our careless. Several studies (Angelakis et al, 2017; Wang et al, 2020) have demonstrated that a close correlation existed between intestinal microbes and productive performance of poultry (Line 61-63). In our previously published paper (Miao et al, 2021), we found significant differences in laying rate only between the control group and S500 group. We speculated that there might be differences in intestinal flora, so the control and S500 group were selected for 16S detection.

Reference:

[1]Angelakis, E. Weight gain by gut microbiota manipulation in productive animals. Microb. Pathog. 2017, 106, 162-170.  https://doi: 10.1016/j.micpath.2016.11.002.

[2]Wang, Y.; Xu, L.; Sun, X.; Wan, X.; Sun, G.; Jiang, R.; Li, W.; Tian, Y.; Liu, X.; Kang, X. Characteristics of the fecal microbiota of high- and low-yield hens and effects of fecal microbiota transplantation on egg production performance. Res. Vet. Sci. 2020, 129, 164-173. https://doi: 10.1016/j.rvsc.2020.01.020.

[3]Miao, S.S.; Zhou, W.T.; Li, H. Y.; Zhu, M. K.; Dong, X. Y.; Zou, X. T. Effects of coated sodium butyrate on production performance, egg quality, serum biochemistry, digestive enzyme activity, and intestinal health of laying hens. Ital. J. Anim. Sci. 2021, 38320, 1452-1461. https://doi:10.1080/1828051x.2021.1960209.

2. Line 43: Based on your definition. The antigen is the foreign particle. So, could you explain the difference between antigens and microbes derived from food in the intestine?

Answer: Thank you for your comments. We apologized for our incorrect expression. The antigens include the food-driven antigens and microbes. (Cabezón, et al, 2013; Wu, et al, 2018). We have rephrased this sentence and cited related references (Line 43-44; Line 360-365). Besides, the reference order has been revised.

[1] Cabezón, R.; Benítez-Ribas, D. Therapeutic potential of tolerogenic dendritic cells in IBD: from animal models to clinical application. Clin. Dev. Immunol. 2013, 2013, 789814. https://doi: 10.1155/2013/789814.

[2] Wu, P.; Tian, L.; Zhou, X.Q.; Jiang, W.D.; Liu, Y.; Jiang, J.; Xie, F.; Kuang, S.Y.; Tang, L.; Tang, W.N.; Yang, J.; Zhang, Y.A.; Shi, H.Q.; Feng, L. Sodium butyrate enhanced physical barrier function referring to Nrf2, JNK and MLCK signaling pathways in the intestine of young grass carp (Ctenopharyngodon idella). Fish. Shellfish. Immunol. 2018, 73, 121-132. https://doi: 10.1016/j.fsi.2017.12.009.

3.Line 72: Why did you choose 52 wks laying hen for your study?Answer: With the increase of age, metabolic disease intensification in the later stage of laying is a bottleneck problem to be solved. The nutritional regulation of intestinal health of laying hens in late laying period is urgent extremely. In this study, the laying rate of huafeng laying hens at 52 weeks is about 73%, which is in the late stage of laying. Therefore, 52 week-old laying hens were selected to explore the nutritional regulation of coated sodium butyrate on intestinal health of laying hens. This condition you have to mention in the materials and method section with citation.

Answer: Thank you for your comments. We have added and cited it in the “Material and Method” in  the revised manuscript according to your suggestion. (Line 78-79)

Round 3

Reviewer 1 Report

The author did touch on all the comments. I hope this format of this manuscript could be published in Animals. So, I recommend accepting this manuscript.